# A New Accelerated Gradient Method Inspired by Continuous-Time Perspective

## Abstract

Nesterov's accelerated method are widely used in problems with machine learning background including deep learning. To give more insight about the acceleration phenomenon, an ordinary differential equation was obtained from Nesterov's accelerated method by taking step sizes approaching zero, and the relationship between Nesterov's method and the differential equation is still of research interest. In this work, we give the precise order of the iterations of Nesterov's accelerated method converging to the solution of derived differential equation as step sizes go to zero. We then present a new accelerated method with higher order. The new method is more stable than ordinary method for large step size and converges faster. We further apply the new method to matrix completion problem and show its better performance through numerical experiments.

## 1 Introduction

Optimization is a core component of statistic and machine learning problems. Recently, gradient-based algorithms are widely used in such optimization problems due to its simplicity and efficiency for large-scale situations. For solving convex optimization problem

$$\min_{x \in \mathbb{R}^d} F(x),$$

where $F(x)$ is convex and sufficiently smooth, a classical first-order method is gradient descent. We assume that $f(x) = \nabla F(x)$ satisfies $L$-Lipschitz condition, that is, there exists constant $L$ such that

$$\|f(x) - f(y)\| \le L\|x - y\|, \qquad \forall x, y.$$

Under these conditions, gradient descent achieves a convergence rate of $\mathcal{O}(n^{-1})$, i.e., $\|F(x_n) - F(x^*)\|$ decreases to zero at a rate of $\mathcal{O}(n^{-1})$, where $x_n$ denotes the $n$th iteration and $x^*$ denotes the minimum point of $F(x)$ in $\mathbb{R}^d$.

Nesterov's accelerated method (Nesterov, 1983) is a more efficient first-order algorithm than gradient descent, of which we will use the following form: starting with $x_0 = x_1$,

$$
\begin{aligned}
y_n &= x_n + \frac{n-3}{n}(x_n - x_{n-1}), \\
x_{n+1} &= y_n - sf(y_n)
\end{aligned}
\tag{1.1}
$$

for $n \ge 1$. It is shown that under abovementioned conditions, Nesterov's accelerated method converges at a rate of $\mathcal{O}(n^{-2})$.

Accelerated gradient method has been successful in training deep and recurrent neural networks (Sutskever et al., 2013) and is widely used in problems with machine learning background to avoid

sophisticated second-order methods (Cotter et al., 2011; Hu et al., 2009; Ji & Ye, 2009). To provide more theorical understanding, an important research topic of Nesterov's accelerated method is to find an explanation of the acceleration. On this topic, Nesterov's method was studied via a continuous-time perspective (Su et al., 2014). They considered a curve $x(t)$, introduced the ansatz $x_n \approx x(n\sqrt{s})$ and substituted it to (1.1). Letting $s \to 0$, they obtained the following differential equation.

$$\ddot{x} + \frac{3}{t}\dot{x} + f(x) = 0. \tag{1.2}$$

The differential equation was used as a tool for analyzing and generalizing Nesterov's scheme. Furthermore, this idea has been studied from different directions. A class of accelerated methods have been generated in continuous-time (Wibisono et al., 2016). ODE (1.2) can also be discretized directly using Runge-Kutta method to achieve acceleration (Zhang et al., 2018).

Although many results have been achieved, the process of obtaining the differential equation (1.2) has not been rigorous, and the method is still time-consuming for large-scale problems. In this work, we give the precise order of the iterations of Nesterov's accelerated method converging to solution of the differential equation (1.2) with initial conditions

$$x(0) = x_0, \ \dot{x}(0) = 0 \tag{1.3}$$

as step size $s$ goes to zero. Inspired from this perspective, we present a new accelerated method to make this convergence faster. As we expected, iterations of the new method are closer to the solution $x(t)$ of differential equation (1.2) than original Nesterov's method. Moreover, we find the new method is more stable than original Nesterov's method when step size is large.

Based on abovementioned observations, we try to take advantage of the new method in more practical problems. We apply the new method to matrix completion problem. We combine the new method with proximal operator (Parikh & Boyd, 2014) into a new algorithm, which we call modified FISTA. We find that the new method performs better than FISTA (Beck & Teboulle, 2009) and acclerated proximal gradient method (Parikh & Boyd, 2014) because it can work with larger step sizes.

This paper is organized as follows. In section 2, we prove that iterations of Nesterov's accelerated method converge to solution of the differential equation (1.2). In section 3, we present a new method to make the convergence faster and show its better stablity through two simple examples. In section 4, we apply the new method to matrix completion problem.

## 2 A strict analysis of the relation between Nesterov's method and its continuous-time limit

We refer to $x(t)$ as the solution of differential equation (1.2) with initial conditions (1.3). Existance and uniqueness of such solutions have been proved (Su et al., 2014). In this section, We give the order of the iterations of Nesterov's accelerated method converging to $x(t)$ as step sizes go to zero.

For convenience, we substitute the first equation in Nesterov's method (1.1) to the second one to get

$$x_{n+1} = x_n + \frac{n-3}{n}(x_n - x_{n-1}) - s \cdot f\left(x_n + \frac{n-3}{n}(x_n - x_{n-1})\right).$$

We write $s = h^2$ and rewrite the above recurrence relation as

$$x_{n+1} = x_n + \frac{n-3}{n}(x_n - x_{n-1}) - h^2 \cdot f\left(x_n + \frac{n-3}{n}(x_n - x_{n-1})\right). \tag{2.1}$$

Inspired by the ansatz $x_n \approx x(n\sqrt{s})$ (Su et al., 2014), we consider the convergence between $x_n$ and $x(nh)$. More precisely, we show that for fixed time $t$, $x_n$ converges to $x(t)$ as $h$ goes to zero, where $n = \frac{t}{h}$.

## 2.1 Truncation error

Firstly, we consider the following 'truncation error':

$$
\begin{aligned}
L[x(t); h] =& x(t+h) - \frac{2t-3h}{t}x(t) + \frac{t-3h}{t}x(t-h) + \\
& h^2 f\left(x(t) + \frac{t-3h}{t}\left(x(t) - x(t-h)\right)\right).
\end{aligned}
\tag{2.2}
$$

(2.2) is obtained from (2.1) by replacing $x_{n+1}$, $x_n$, $x_{n-1}$ with $x(t+h)$, $x(t)$, $x(t-h)$ and substituting the relation $n = \frac{t}{h}$. Our first result is the order of truncation error $L[x(t); h]$.

Theorem 1. Assume $f$ satisfies $L$-Lipschitz condition, and solution $x(t)$ of the derived differential equation (1.2) has a continuous third derivative. For fixed time $t$, the truncation error (2.2) satisfies

$$
L[x(t); h] = \mathcal{O}(h^3).
$$

Theorem 1 shows the size of error caused by a single iteration when the starting point is just on $x(t)$. Then we have to add up these errors to prove the convergence proporty we need.

## 2.2 Convergence theorem

We now come to the convergence theorem. In this theorem, we give the precise order of the iterations of Nesterov's method converging to solution of the derived differential equation.

Theorem 2. Under conditions in Theorem 1, for fixed time $t$, $x_{t/h}$ converges to $x(t)$ as $h$ goes to zero at a rate of $\mathcal{O}(h \ln \frac{1}{h})$ if $x_0 = x(0)$ and $x_1 = x(h)$.

Theorem 2 coincides with derivation of ODE (1.2) (Su et al., 2014).

# 3 New accelerated method

## 3.1 Derivation of the new method and analysis of truncation error

Inspired from the continuous-time perspective and our proof of the convergence from iterations of Nesterov's method to its continuous-time limit, we present a new method to make this convergence faster. Precisely, the new method has a higher truncation order.

We need one more step in our scheme than in Nesterov's method to achieve higher truncation order in the following analysis, so we consider a recurrence relation with form

$$
\sum_{i=1}^{4}\left(\alpha_i + \frac{\beta_i}{n} + \frac{\gamma_i}{n^2}\right)x_{n+2-i} = -sf\left(x_n + \frac{n-3}{n}(x_n - x_{n-1})\right),
\tag{3.1}
$$

where $\{\alpha_i\}$, $\{\beta_i\}$ and $\{\gamma_i\}$ are to be determined.

Now we expand $x(t-h)$ to first order. Calculation shows that

$$f\left(x(t)+\frac{t-3h}{t}(x(t)-x(t-h))\right)= -hx^{(3)}(t)-\left(\frac{3h}{t}+1\right)x^{(2)}(t)$$
$$+\left(\frac{3h}{t^2}-\frac{3}{t}\right)x^{(1)}(t)+\mathcal{O}(h^2).$$

Substitute this expansion to truncation error

$$L[x(t);h]=\sum_{i=1}^{4}\left(\alpha_i+\frac{\beta_i h}{t}+\frac{\gamma_i h^2}{t^2}\right)x(t+(2-i)h)$$
$$+h^2 f\left(x(t)+\frac{t-3h}{t}(x(t)-x(t-h))\right),$$

and choose parameters appropriately to eliminate low-order terms, we get the following recurrence relation

$$x_{n+1}=\frac{10n^2+9n+6}{4n^2+8n}x_n-\frac{4n^2+3}{2n^2+4n}x_{n-1}+\frac{2n-1}{4n+8}x_{n-2}$$
$$-\frac{n}{2n+4}sf\left(\frac{2n-3}{n}x_n-\frac{n-3}{n}x_{n-1}\right). \tag{3.2}$$

Here we rewrite this scheme as Algorithm 1.

---

**Algorithm 1** The new method (3.2)

---

Input: step size $s$
Initial value: $\boldsymbol{X}_2=\boldsymbol{X}_1=\boldsymbol{X}_0$.
$(\boldsymbol{k-1})$th iteration $(k\geq 2)$. Compute

$$\boldsymbol{Y}_k=\frac{10k^2+9k+6}{4k^2+8k}\boldsymbol{X}_k-\frac{4k^2+3}{2k^2+4k}\boldsymbol{X}_{k-1}+\frac{2k-1}{4k+8}\boldsymbol{X}_{k-2},$$
$$\boldsymbol{Z}_k=\frac{2k-3}{k}\boldsymbol{X}_k-\frac{k-3}{k}\boldsymbol{X}_{k-1},$$
$$\boldsymbol{X}_{k+1}=\boldsymbol{X}-\left(\boldsymbol{Y}_k-\frac{ks}{2k+4}f(\boldsymbol{Z}_k)\right).$$

---

For truncation order of this new method, we have the following theorem. The abovementioned procedure is presented in Appendix A.4 detailedly, as proof of Theorem 3.

**Theorem 3.** If $f$ has continuous second order derivative, the first and second derivative are bounded, and $x(t)$ has continuous fourth derivative, then for fixed $t$, truncation error of (3.2) satisfies

$$L[x(t_n);h]=\mathcal{O}(h^4).$$

The convergence of the new method and $x(t)$ can be proved similar to Theorem 2.

### 3.2 Advantage of the new method

Since the new method has a truncation error of higher order than original Neaterov's method, the iterations of the new method converge to the differential equation (1.2) when those of original

Nesterov's method diverge. In another word, the new method is more stable for large step size. We present two numerical results in Figure 1 to confirm it.

Quadratic. $F(\boldsymbol{x}) = \boldsymbol{x}^\mathsf{T} \boldsymbol{A} \boldsymbol{x}$ is a strongly convex function, in which $\mathbf{x} \in \mathbb{R}^\mathbf{2}$ and $\boldsymbol{A}$ is a $2 \times 2$ matrix.

Linear regression. $F(\boldsymbol{x}) = \sum_{i=1}^{n} (y_i - \boldsymbol{w_i}^\mathsf{T} \boldsymbol{x})^2$, where $n$ is the number of samples and $(\boldsymbol{w_i}, y_i)$ is the $i$th sample.

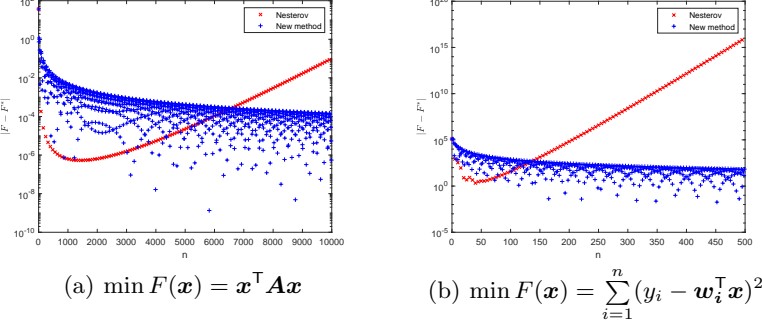

(a) $\min F(\boldsymbol{x}) = \boldsymbol{x}^\mathsf{T} \boldsymbol{A} \boldsymbol{x}$       (b) $\min F(\boldsymbol{x}) = \sum_{i=1}^{n} (y_i - \boldsymbol{w_i}^\mathsf{T} \boldsymbol{x})^2$

Figure 1: Iterations of original Nesterov's method (Nesterov) and the new method (New method) for quodratic and linear regression objective function. Y-axis represents the gap $|F(x_n) - F(x^*)|$. In Figure 1(a), step size $s = 0.03705$; in Figure 1(b), step size $s = 0.00565$.

In these examples, iterations of the new method converge to the minimum point, while those of original Nesterov's method diverge, which confirms that the new method is more stable for large step size.

### 3.3 Absolute stability of Nesterov's method and the new method

In this subsection, we explain the better stability of the new method with absolute stability theory. Firstly, recall the scheme of our new method

$$x_{n+1} = \frac{10n^2 + 9n + 6}{4n^2 + 8n} x_n - \frac{4n^2 + 3}{2n^2 + 4n} x_{n-1} + \frac{2n - 1}{4n + 8} x_{n-2}$$
$$- \frac{n}{2n + 4} sf \left( \frac{2n - 3}{n} x_n - \frac{n - 3}{n} x_{n-1} \right).$$

We use linear approximation

$$f \left( x_n + \frac{n - 3}{n} (x_n - x_{n-1}) \right) = \nabla F \left( x_n + \frac{n - 3}{n} (x_n - x_{n-1}) \right) \approx \nabla^2 F \cdot \left( x_n + \frac{n - 3}{n} (x_n - x_{n-1}) \right),$$

and the characteristic equation of this finite scheme is approximately

$$\lambda^3 - \left( \frac{10n^2 + 9n + 6}{4n^2 + 8n} - s \cdot \nabla^2 F \cdot \frac{2n^2 - 3n}{2n^2 + 4n} \right) \lambda^2 + \left( \frac{4n^2 + 3}{2n^2 + 4n} - s \cdot \nabla^2 F \cdot \frac{n^2 - 3n}{2n^2 + 4n} \right) \lambda - \frac{2n - 1}{4n + 8} = 0.$$

For large $n$, we can ignore the high order terms and the characteristic equation becomes

$$\lambda^3 - \left( \frac{5}{2} - s \cdot \nabla^2 F \cdot \right) \lambda^2 + \left( 2 - \frac{s}{2} \cdot \nabla^2 F \right) \lambda - \frac{1}{2} = 0.$$

According to the absolute stability theory, the numerical stability of Nesterov's scheme with respect to accumulated roundoff error is equivalent to this: all the roots of the characteristic equation lie in the unit circle (Leader, 2004). Noticing that the left hand of the equation can be factorized to

$$\left(\lambda - \frac{1}{2}\right)\left(\lambda^2 - (2 - s \cdot \nabla^2 F)\lambda + 1\right),$$

the largest modulu of the roots is 1 when $0 \leq s \cdot \nabla^2 F \leq 4$, and the absolutely stable region of the new method is $s \cdot \nabla^2 F \in [0, 4]$.

When $s \cdot \nabla^2 F$ lies in the absoletely stable region, the related theory guarantees that the error caused by every iteration will not be magnified as the iteration number increases. To make the analysis more precise, we should consider the difference of the scheme between iterations caused by different $n$. We define the transfer matrix

$$P_n = \begin{pmatrix} \left(\frac{10n^2+9n+6}{4n^2+8n} - s \cdot \nabla^2 F \cdot \frac{2n^2-3n}{2n^2+4n}\right) & -\left(\frac{4n^2+3}{2n^2+4n} - s \cdot \nabla^2 F \cdot \frac{n^2-3n}{2n^2+4n}\right) & \frac{2n-1}{4n+8} \\ 1 & 0 & 0 \\ 0 & 1 & 0 \end{pmatrix}$$

and $Q_n = P_n P_{n-1} \cdots P_1$. Error analysis shows that if the largest modulu of eigenvalues of $Q_n$ goes to zero, then error caused by iterations will be eliminated as the iteration number increases. Figure 2 presents the largest module of eigenvalues of $Q_n$ for different values of $s \cdot \nabla^2 F$. From the experiment we can see that the above condition is satisfied.

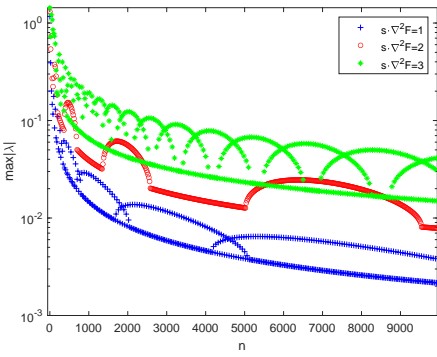

Figure 2: The largest modulu of eigenvalues of $Q_n$, where $s \cdot \nabla^2 F$ is chosen to be 1, 2 and 3.

We then apply the same method to Nesterov's method discussed in (Su et al., 2014) and conclude that the absolutely stable region of Nesterov's method is $[0, \frac{4}{3}]$.

According to the above analysis, the absolutely stable region of the new method is four times as large as Nesterov's method, so the new method is more stable, and we can choose larger step sizes to achieve faster convergence.

## 4  Application to matrix completion problem: modified FISTA

Our theory and numerical results show that the new method is more stable than original Nestrov's method. So we can choose larger step size for new method and convergence to the optimal solution

can be faster, compared with original Nesterov's method. In this section we apply the new method to matrix completion problem. We present a new algorithm which can be viewed as a modification of the well-konwn fast iterative shrinkage-thresholding algorithm (FISTA) (Beck & Teboulle, 2009). The performance of modified FISTA can also confirm the advantage of the new method.

For matrix completion problem there exists a 'true' low rank matrix $\boldsymbol{M}$. We are given some entries of $\boldsymbol{M}$ and asked to fill missing entries. There have been various algorithms to solve such problem (Candès & Recht, 2009; Keshavan et al., 2010). Besides, it is proposed that matrix completion can be transformed to the following unconstrained optimization problem (Mazumder et al., 2010)

$$\min F(\boldsymbol{X}) = \frac{1}{2}\|\boldsymbol{X}_{obs} - \boldsymbol{M}_{obs}\|^2 + \lambda\|\boldsymbol{X}\|_*. \tag{4.1}$$

Notice that $F(\boldsymbol{X})$ is composed of a smooth term and a non-smooth term, so gradient-based algorithms cannot be used directly. Proximal gradient algorithms (Parikh & Boyd, 2014) are widely used in such composite optimization problems, and fast iterative shrinkage-thresholding algorithm (FISTA) is a successful algorithm. Moreover, FISTA has been extended to matrix completion case (Ji & Ye, 2009). For convenience, we set $G(\boldsymbol{X}) = \frac{1}{2}\|\boldsymbol{X}_{obs} - \boldsymbol{M}_{obs}\|^2$, $H(\boldsymbol{X}) = \lambda\|\boldsymbol{X}\|_*$, and $g(\boldsymbol{X}) = \nabla G(\boldsymbol{X})$.

The idea of FISTA builds on Nesterov's method. We also apply acclerated proximal gradient method (Parikh & Boyd, 2014) for our numerical experiment, which is composed of Nesterov's method and proximal gradient descent. These two algorithms are presented in Appendix A.5. We find the performances of them are similar in our experiments.

Our contribution is the third method (Algorithm 2), the new method (3.2) combined with proximal operator, which we call modified FISTA.

---

**Algorithm 2** Modified FISTA

---

Input: step size $s$
Initial value: $\boldsymbol{X}_2 = \boldsymbol{X}_1 = \boldsymbol{X}_0 \in \boldsymbol{M}_{100}$.
$(\boldsymbol{k-1})$th iteration ($k \geq 2$). Compute

$$\boldsymbol{Y}_k = \frac{10k^2 + 9k + 6}{4k^2 + 8k}\boldsymbol{X}_k - \frac{4k^2 + 3}{2k^2 + 4k}\boldsymbol{X}_{k-1} + \frac{2k-1}{4k+8}\boldsymbol{X}_{k-2},$$

$$\boldsymbol{Z}_k = \frac{2k-3}{k}\boldsymbol{X}_k - \frac{k-3}{k}\boldsymbol{X}_{k-1},$$

$$\boldsymbol{X}_{k+1} = \arg\min_{\boldsymbol{X}} \left\{ \frac{1}{2} \cdot \frac{2k+4}{ks} \left\| \boldsymbol{X} - \left( \boldsymbol{Y}_k - \frac{ks}{2k+4}g(\boldsymbol{Z}_k) \right) \right\|^2 + \lambda\|\boldsymbol{X}\|_* \right\}.$$

---

Notice that the minimizing problems in interations of above three algorithms can be solved directly by singular value decomposition (Cai & Candès, 2010). We take experiments on a simulated data set. We use fixed step sizes in the above three algorithms, and the performances are presented in Figure 3. We find empirically that for all methods, convergence is faster when step size is larger, so we choose the largest step sizes for all methods to compare their fastest convergence speed. Through experiments, we find the largest step size that makes modified FISTA convergent is 4.1 (accurate to one decimal place), while those for the first two algorithms are both 1.3. We also compare performances of the three methods with step sizes reduced from the largest in equal proportion. We find that when step sizes are chosen to be the largest or reduced from the largest in equal proportion (80%, 50%, 10%), convergence of modified FISTA is faster than the other two methods.

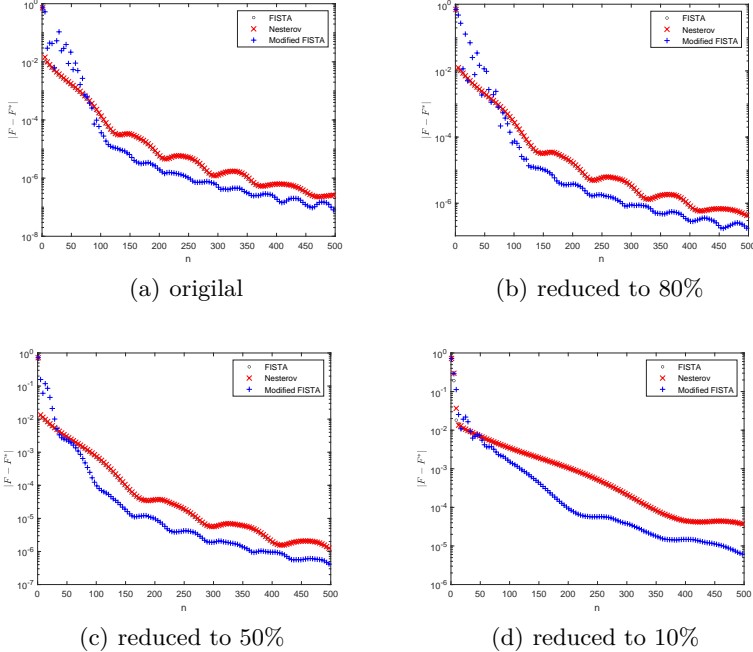

Figure 3: Iterations of FISTA, accelerated proximal gradient descent (Nesterov) and modified FISTA (Modified FISTA) for matrix completion objective function. Y-axis represents the gap $|F(x_n) - F(x^*)|$. In Figure 3(a), step size is 1.3 for FISTA and accelerated proximal gradient descent, and 4.1 for modified FISTA. In the other three figures, step sizes are reduced from 1.3 and 4.1 in the proportion marked below the figures.

We also combine the three methods with backtracking (Beck & Teboulle, 2009) to choose step sizes automatically. We present modified FISTA with backtracking as Algorithm 3, and the other two algorithms are similar.

Performances of the three algorithms with backtracking on abovementioned data set are presented in Figure 4. Convergence of modified FISTA is faster than the other two methods. Moreover, we find that the final step size of modified FISTA is larger.

## 5 Discussion

In this paper we prove that iterations of Nesterov's accelerated method converge to solution of the derived differential equation as step sizes go to zero. We present a new accelerated method to make this convergence faster. We use numerical results to show that the new method is more stable, especially for large step sizes, and explan it using the order of truncation error. We then apply the new method to matrix completion problem and present a new algorithm which we call modified FISTA. Numerical experiments show that modified FISTA performs better than existing algorithms based on Nesterov's acceleration because it can work with larger step sizes. We will also combine our new method with stochastic gradient-based algorithms and apply the new method to deep networks in the future.

---

**Algorithm 3** Modified FISTA with backtracking

---

Input: some $\beta < 1$

Initial value. $\boldsymbol{X}_2 = \boldsymbol{X}_1 = \boldsymbol{X}_0 \in \boldsymbol{M}_{100}$, step size $s_2$.

$(\boldsymbol{k-1})$th iteration $(k \geq 2)$.

$$\boldsymbol{Y}_k = \frac{10k^2 + 9k + 6}{4k^2 + 8k}\boldsymbol{X}_k - \frac{4k^2 + 3}{2k^2 + 4k}\boldsymbol{X}_{k-1} + \frac{2k-1}{4k+8}\boldsymbol{X}_{k-2},$$
$$\boldsymbol{Z}_k = \frac{2k-3}{k}\boldsymbol{X}_k - \frac{k-3}{k}\boldsymbol{X}_{k-1}.$$

Find the smallest positive integer $i_{k+1}$ such that with $s = \beta^{i_{k+1}}s_k$

$$F(\widetilde{\boldsymbol{X}}) < F(\boldsymbol{Y}_k) + \left\langle \widetilde{\boldsymbol{X}} - \boldsymbol{Y}_k, g(\boldsymbol{Z}_k)\right\rangle + \frac{1}{2}\cdot\frac{2k+4}{ks}\|\widetilde{\boldsymbol{X}} - \boldsymbol{Y}_k\|^2,$$

where

$$\widetilde{\boldsymbol{X}} = \arg\min_{\boldsymbol{X}}\left\{\frac{1}{2}\cdot\frac{2k+4}{ks}\left\|\boldsymbol{X} - \left(\boldsymbol{Y}_k - \frac{ks}{2k+4}g(\boldsymbol{Z}_k)\right)\right\|^2 + \lambda\|\boldsymbol{X}\|_*\right\}.$$

Set $s_{k+1} = \beta^{i_{k+1}}s_k$ and compute

$$\boldsymbol{X}_{k+1} = \widetilde{\boldsymbol{X}}.$$

---

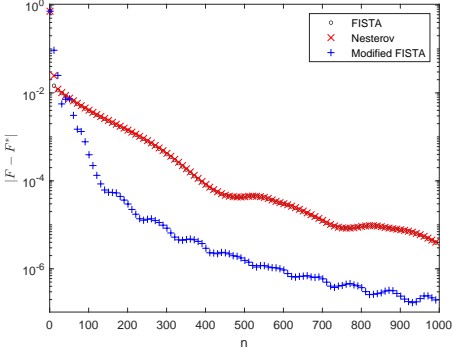

Figure 4: Iterations of FISTA, accelerated proximal gradient descent (Nesterov) and modified FISTA (Modified FISTA) for matrix completion objective function. Y-axis represents the gap $|F(x_n) - F(x^*)|$. Step sizes are chosen by backtracking.

Our work shows that for an accelerated gradient method, the rate at which it converges to the derived differential equation is possibly related to its property as an optimization algorithm. We think this work suggests that more consideration should be given to the corresponding differential equations when studying optimization algorithms.

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

## A Appendix

### A.1 Proof of Theorem 1

Theorem 1. Assume $f$ satisfies $L$-Lipschitz condition, and solution $x(t)$ of the derived differential equation (1.2) has a continuous third derivative. For fixed time $t$, the truncation error (2.2) satisfies

$$L[x(t); h] = \mathcal{O}(h^3). \tag{A.1}$$

Proof. Notice that

$$x(t - h) = x(t) + \mathcal{O}(h).$$

Substitute this equation to the last term of $L[x(t); h]$ to get

$$f\left(x(t) + \frac{t - 3h}{t}(x(t) - x(t - h))\right) = f\left(x(t) + \frac{t - 3h}{t} \cdot \mathcal{O}(h)\right).$$

Since $f$ satisfies $L$-Lipschitz condition, we know

$$f\left(x(t) + \frac{t - 3h}{t}(x(t) - x(t - h))\right) = f(x(t)) + \mathcal{O}(h)$$

$$= -\ddot{x}(t) - \frac{3}{t}\dot{x}(t) + \mathcal{O}(h).$$

To get the second equality, we substitute the differential equation (1.2). Then we expend the first and third terms of $L[x(t); h]$ to third order to get

$$x(t + h) = x(t) + hx^{(1)}(t) + \frac{h^2}{2}x^{(2)}(t) + \mathcal{O}(h^3),$$

$$x(t - h) = x(t) - hx^{(1)}(t) + \frac{h^2}{2}x^{(2)}(t) + \mathcal{O}(h^3).$$

Substitute these three equations to (2.2), we have

$$L[x(t); h] = \mathcal{O}(h^3). \qquad \square$$

Remark 1. (A.1) can also be written as

$$|L[x(t); h]| \le M_1 h^3,$$

where $M_1$ depends on $\sup_{s \le t} |x^{(1)}(s)|$ and $\sup_{s \le t} |x^{(3)}(s)|$.

Remark 2. Theorem 1 deals with the problem for fixed time $t$. To finish the proof of the convergence, we have to consider the situation that $t_n = nh$, where $n \ge 1$ is fixed.

We set a fixed time $t_0$ and assume that $t_n = nh < t_0$. Since $x(t)$ has a continuous third derivative, $x(t)$ and its first to third derivative are bounded in $[0, t_0]$. We replace time $t$ in the above proof by $t_n$ and expend the terms of (2.2). Now the term

$$-\frac{3h^3}{2t_n}x^{(2)}(t_n)$$

obtained from the expansion of $x(t_{n-1})$ cannot be viewed as $\mathcal{O}(h^3)$, but there exists $M_2 > 0$ such that

$$\left| -\frac{3h^3}{2t_n}x^{(2)}(t_n) \right| \le M_2 \frac{h^2}{n}.$$

As a consequence, we have

$$|L[x(t_n); h]| \le M_1 h^3 + M_2 \frac{h^2}{n}, \tag{A.2}$$

where $M_1$ and $M_2$ rely on $t_0$.

## A.2 Two lemmas for Theorem 2

For the proof of Theorem 2, we need the following two lemmas.

Lemma 1. (Holte, 2009) For constant $\alpha$, $\beta > 0$ and positive sequence $\{\eta_n\}_{n \geq 0}$ satisfying

$$\eta_n \leq \beta + \alpha \sum_{i=0}^{n-1} \eta_i, \quad \forall n > 0,$$

the following inequality holds

$$\eta_n \leq e^{\alpha n}(\beta + \alpha \eta_0).$$

The above lemma is a classic result and refered to as discrete Gronwall inequality.

Lemma 2. We define matrices $\boldsymbol{C}_n$ and $\boldsymbol{D}_{n,l}$ as

$$\boldsymbol{C}_n = \begin{pmatrix} \frac{2n-1}{n+1} & -\frac{n-2}{n+1} \\ 1 & 0 \end{pmatrix},$$

$$\boldsymbol{D}_{n,l} = \boldsymbol{C}_n \boldsymbol{C}_{n-1} \cdots \boldsymbol{C}_{n-l+1},$$

where $n \geq 0$ and $0 < l \leq n+1$. In addition, we set $\boldsymbol{D}_{n,0} = \boldsymbol{I}_2$. Then there exist positive constants $M$, $M_3$ such that for all $n$, the following two inequalities hold, where the matrix norm is 2-norm.

$$\sup_{0 \leq l \leq n+1} \|\boldsymbol{D}_{n,l}\| \leq Mn,$$

$$\boldsymbol{D}_{n,n+1} \leq M_3. \tag{A.3}$$

Proof. Since

$$\boldsymbol{C}_2 = \begin{pmatrix} 1 & 0 \\ 1 & 0 \end{pmatrix},$$

we notice that when $n \geq 2$,

$$\boldsymbol{D}_{n,n-1} = \begin{pmatrix} 1 & 0 \\ 1 & 0 \end{pmatrix}, \quad \boldsymbol{D}_{n,n} = \begin{pmatrix} \frac{1}{2} & \frac{1}{2} \\ \frac{1}{2} & \frac{1}{2} \end{pmatrix}, \quad \boldsymbol{D}_{n,n+1} = \begin{pmatrix} 0 & 1 \\ 0 & 1 \end{pmatrix},$$

having nothing to do with the value of $n$. So it is obvious that there exists $M_3$ to make (A.3) holds and $M_4 > 0$ such that for all $n < 2$ or $n \geq 2$, $l > n-2$ or $l = 0$,

$$\|\boldsymbol{D}_{n,l}\| \leq M_4 n. \tag{A.4}$$

Then we consider the condition when $n \geq 2$, $0 < l \leq n-2$. Notice that

$$\boldsymbol{C}_n = \begin{pmatrix} 1 & 1 \\ 1 & 0 \end{pmatrix} \begin{pmatrix} 1 & 1 \\ 0 & \frac{n-2}{n+1} \end{pmatrix} \begin{pmatrix} 1 & 1 \\ 1 & 0 \end{pmatrix}^{-1}.$$

For convenience, we write

$$\boldsymbol{P} = \begin{pmatrix} 1 & 1 \\ 1 & 0 \end{pmatrix}.$$

Assume we have already got

$$\boldsymbol{D}_{n,l} = \boldsymbol{P} \begin{pmatrix} 1 & a_{n,l} \\ 0 & b_{n,l} \end{pmatrix} \boldsymbol{P}^{-1}$$

satisfying

$$0 < a_{n,l} \leq l, \quad 0 < b_{n,l} \leq 1,$$

then since
$$\boldsymbol{D}_{n,l+1} = \boldsymbol{D}_{n,l}\boldsymbol{C}_{n-l},$$
and $0 \le \frac{n-l-2}{n-l+1} < 1$, $\boldsymbol{D}_{n,l+1}$ has the same form
$$\boldsymbol{D}_{n,l+1} = \boldsymbol{P}\begin{pmatrix} 1 & a_{n,l+1} \\ 0 & b_{n,l+1} \end{pmatrix}\boldsymbol{P}^{-1},$$
satisfying
$$0 < a_{n,l+1} \le l+1, \qquad 0 < b_{n,l} \le 1.$$
Then for fixed $n$, induce from $l = 1$, we get
$$\boldsymbol{D}_{n,l} = \boldsymbol{P}\widetilde{\boldsymbol{D}}_{n,l}\boldsymbol{P}^{-1} \triangleq \boldsymbol{P}\begin{pmatrix} 1 & a_{n,l} \\ 0 & b_{n,l} \end{pmatrix}\boldsymbol{P}^{-1},$$
satisfying
$$0 < a_{n,l} \le l \le n, \qquad 0 < b_{n,l} \le 1, \tag{A.5}$$
for all $n \ge 2$, $0 < l \le n-2$. Then we can estimate $\|\boldsymbol{D}_{n,l}\|$. Notice that
$$\widetilde{\boldsymbol{D}}_{n,l}\widetilde{\boldsymbol{D}}_{n,l}^{\mathsf{T}} = \begin{pmatrix} 1 + a_{n,l}^2 & a_{n,l}b_{n,l} \\ a_{n,l}b_{n,l} & a_{n,l}^2 \end{pmatrix}.$$
The eigenvalues of this matrix are
$$\lambda_{1,2} = \frac{1 + a_{n,l}^2 + b_{n,l}^2 \pm \sqrt{(1 + a_{n,l}^2 + b_{n,l}^2)^2 - 4b^4}}{2}.$$
Combining this representation with (A.5), we get the estimation
$$\|\widetilde{\boldsymbol{D}}_{n,l}\| = \sqrt{\max\{|\lambda_1|, |\lambda_2|\}} \le \sqrt{1 + a_{n,l}^2 + b_{n,l}^2} \le n+2.$$
So there exists $M_5 > 0$, such that for all $n \ge 2$, $0 < l \le n-2$, inequality
$$\|\boldsymbol{D}_{n,l}\| \le M_5 n \tag{A.6}$$
holds. Combining (A.4) with (A.6), we finish the proof. $\qquad\square$

### A.3 Proof of Theorem 2

Theorem 2. Under conditions in Theorem 1, for fixed time $t$, $x_{t/h}$ converges to $x(t)$ as $h$ goes to zero at a rate of $\mathcal{O}(h \ln \frac{1}{h})$ if $x_0 = x(0)$ and $x_1 = x(h)$.

Proof. In this proof, we first calculate the error caused by a single iteration, which can be divided into an accumulation term and a truncation term. Then we use the estimation given by Theorem 1 and apply discrete Gronwall inequality to prove the convergence.

Recall the recurrence relation
$$x_{n+1} = x_n + \frac{n-3}{n}(x_n - x_{n-1}) - h^2 \cdot f\left(x_n + \frac{n-3}{n}(x_n - x_{n-1})\right)$$
and the definition of truncation error
$$x(t_{n+1}) = x(t_n) + \frac{n-3}{n}(x(t_n) - x(t_{n-1})) - h^2 f\left(x(t_n) + \frac{n-3}{n}(x(t_n) - x(t_{n-1}))\right) + L[x(t_n); h],$$

where $t_n = nh$.

Subtract the above two equations, and introduce overall error

$$e_n = x(t_n) - x_n,$$

we have

$$e_{n+1} = \frac{2n-3}{n}e_n - \frac{n-3}{n}e_{n-1} - h^2 b_{n-1} + L[x(t_n); h],$$

which can also be written as

$$e_{n+2} - \frac{2n-1}{n+1}e_{n+1} + \frac{n-2}{n+1}e_n = -h^2 b_n + L[x(t_{n+1}); h], \tag{A.7}$$

where

$$b_n = f\left(\frac{2n-1}{n+1}x_{n+1} - \frac{n-2}{n+1}x_n\right) - f\left(\frac{2n-1}{n+1}x(t_{n+1}) - \frac{n-2}{n+1}x(t_n)\right). \tag{A.8}$$

We will also use the notation

$$b_n^* = -\frac{e_{n+2} - \frac{2n-1}{n+1}e_{n+1} + \frac{n-2}{n+1}e_n}{h^2}.$$

Then we rewrite (A.7) into a form that is convenient for recurrence. We set

$$\boldsymbol{E}_n = \begin{pmatrix} e_{n+1} \\ e_n \end{pmatrix}, \quad \boldsymbol{C}_n = \begin{pmatrix} \frac{2n-1}{n+1} & -\frac{n-2}{n+1} \\ 1 & 0 \end{pmatrix}, \quad \boldsymbol{B}_n = \begin{pmatrix} -h^2 b_n^* \\ 0 \end{pmatrix}.$$

Then (A.7) can be written as

$$\boldsymbol{E}_{n+1} = \boldsymbol{C}_n \boldsymbol{E}_n + \boldsymbol{B}_n.$$

By recursive method, we have

$$\boldsymbol{E}_n = \boldsymbol{C}_{n-1}\cdots\boldsymbol{C}_0\boldsymbol{E}_0 + \sum_{l=1}^{n}\boldsymbol{C}_{n-1}\cdots\boldsymbol{C}_{n-l+1}\boldsymbol{B}_{n-l}.$$

With the notations introduced in Lemma 2, this equation can be written as

$$\boldsymbol{E}_n = \boldsymbol{D}_{n-1,n}\boldsymbol{E}_0 + \sum_{l=1}^{n}\boldsymbol{D}_{n-1,l-1}\boldsymbol{B}_{n-l}. \tag{A.9}$$

Now we need to estimate $\|\boldsymbol{B}_n\|$. Since $f$ satisfies $L$-Lipschitz condition, from (A.8) we have

$$|b_n| \le L\left(\frac{2n-1}{n+1}|e_{n+1}| + \frac{n-2}{n+1}|e_n|\right) \le L\left(2|e_{n+1}| + |e_n|\right) \le 3L\|\boldsymbol{E}_n\|.$$

and

$$\|\boldsymbol{B}_n\| \le 3h^2 L\|\boldsymbol{E}_n\| + L[x(t_{n+1}); h]. \tag{A.10}$$

Take norm on both sides of (A.9) and substitute (A.10) and conclusion of Lemma 2, we have the following estimation

$$\|\boldsymbol{E}_n\| \le M_3\|\boldsymbol{E}_0\| + M(n-1)\sum_{l=0}^{n-1}\left(3h^2 L\|\boldsymbol{E}_l\| + L[x(t_{l+1}); h]\right)$$

$$\le M_3\|\boldsymbol{E}_0\| + 3Mnh^2 L\sum_{l=0}^{n-1}\|\boldsymbol{E}_l\| + Mn\sum_{l=0}^{n-1}L[x(t_{l+1}); h]. \tag{A.11}$$

Now we deal with truncation errors. Recall (A.2) in remark of Theorem 1

$$|L[x(t_l); h]| \le M_1 h^3 + M_2 \frac{h^2}{l}.$$

Take sum to obtain

$$\sum_{l=0}^{n-1} |L[x(t_{l+1}); h]| \le n M_1 h^3 + M_2 h^2 \sum_{l=0}^{n-1} \frac{1}{l+1}. \tag{A.12}$$

Notice the classic inequality

$$\sum_{i=1}^{n} \frac{1}{i} \le \ln n + M_e,$$

where $M_e$ refers to a positive constant. Substitute it to (A.12), we have

$$\sum_{l=0}^{n-1} |L[x(t_{l+1}); h]| \le n M_1 h^3 + M_2 h^2 (\ln n + M_e).$$

Substitute this inequality to (A.11), we get a control of $\|\boldsymbol{E}_n\|$

$$\|\boldsymbol{E}_n\| \le M_3 \|\boldsymbol{E}_0\| + 3Mnh^2 L \sum_{l=0}^{n-1} \|\boldsymbol{E}_l\| + MM_1 n^2 h^3 + MM_2 M_e n h^2 + MM_2 n h^2 \ln n$$

Using discrete Gronwall inequality, we have

$$\|\boldsymbol{E}_n\| \le e^{3Mn^2 h^2 L} \left( M_3 \|\boldsymbol{E}_0\| + MM_1 n^2 h^3 + MM_2 M_e n h^2 + MM_2 n h^2 \ln n + 3Mnh^2 L \|\boldsymbol{E}_0\| \right).$$

Then for fixed $t$, we choose $n = \frac{t}{h}$ to get

$$\|\boldsymbol{E}_{t/h}\| \le e^{3Mt^2 L} \left( (M_3 + 3MthL) \|\boldsymbol{E}_0\| + (MM_1 t^2 + MM_2 M_e t) h + MM_2 th \ln \frac{t}{h} \right).$$

Notice that

$$\lim_{h \to 0} h \ln \frac{t}{h} = 0,$$

so if $\boldsymbol{E}_0 = \boldsymbol{0}$, then the vector form of overall error $\boldsymbol{E}_{t/h}$ satisfies

$$\lim_{h \to 0} \|\boldsymbol{E}_{t/h}\| = 0. \qquad \square$$

### A.4  Proof of Theorem 3

Theorem 3. If $f$ has continuous second order derivative, the first and second derivative are bounded, and $x(t)$ has continuous fourth derivative, then for fixed $t$, truncation error of (3.2) satisfies

$$L[x(t); h] = \mathcal{O}(h^4).$$

Proof. Recall the proof of Throrem 1. Now we expand $x(t - h)$ to first order

$$x(t - h) = x(t) + hx^{(1)}(t) + \mathcal{O}(h^2).$$

Then we have

$$
f\left(x(t)+\frac{t-3h}{t}(x(t)-x(t-h))\right) = f\left(x(t)+\left(1-\frac{3h}{t}\right)(hx^{(1)}(t)+\mathcal{O}(h^2))\right)
$$
$$
= f\left(x(t)+hx^{(1)}(t)+\mathcal{O}(h^2)\right)
$$
$$
= f\left(x(t)+hx^{(1)}(t)\right)+\mathcal{O}(h^2).
$$

We now expand $f$:

$$
f\left(x(t)+\frac{t-3h}{t}(x(t)-x(t-h))\right) = f(x(t))+hx^{(1)}(t)f^{(1)}(x(t))+\mathcal{O}(h^2).
$$

To do this, we need $f$ has continuous second derivative and the second derivative is bounded. Take derivetive on both sides of differential equation

$$
\ddot{x}+\frac{3}{t}\dot{x}+f(x)=0,
$$

we have

$$
(f(x(t)))' = -x^{(3)}(t)-\frac{3}{t}x^{(2)}(t)+\frac{3}{t^2}x^{(1)}(t).
$$

So

$$
f\left(x(t)+\frac{t-3h}{t}(x(t)-x(t-h))\right) = -hx^{(3)}(t)-\left(\frac{3h}{t}+1\right)x^{(2)}(t)
$$
$$
+\left(\frac{3h}{t^2}-\frac{3}{t}\right)x^{(1)}(t)+\mathcal{O}(h^2). \tag{A.13}
$$

Expand $x(t+h)$, $x(t-h)$, $x(t-2h)$ to the third order, we have

$$
\left(\alpha_1+\frac{\beta_1 h}{t}+\frac{\gamma_1 h^2}{t^2}\right)x(t+h) = \left(\alpha_1+\frac{\beta_1 h}{t}+\frac{\gamma_1 h^2}{t^2}\right)
$$
$$
\left[x(t)+hx^{(1)}(t)+\frac{h^2}{2}x^{(2)}(t)+\frac{h^3}{6}x^{(3)}(t)+O(h^4)\right],
$$
$$
\left(\alpha_3+\frac{\beta_3 h}{t}+\frac{\gamma_3 h^2}{t^2}\right)x(t-h) = \left(\alpha_3+\frac{\beta_3 h}{t}+\frac{\gamma_3 h^2}{t^2}\right)
$$
$$
\left[x(t)-hx^{(1)}(t)+\frac{h^2}{2}x^{(2)}(t)-\frac{h^3}{6}x^{(3)}(t)+O(h^4)\right],
$$
$$
\left(\alpha_4+\frac{\beta_4 h}{t}+\frac{\gamma_4 h^2}{t^2}\right)x(t-2h) = \left(\alpha_4+\frac{\beta_4 h}{t}+\frac{\gamma_4 h^2}{t^2}\right)
$$
$$
\left[x(t)-2hx^{(1)}(t)+2h^2 x^{(2)}(t)-\frac{4h^3}{3}x^{(3)}(t)+O(h^4)\right].
$$

Substitute these three equations and (A.13) to truncation error of recurrence relation (3.1)

$$
L[x(t);h] = \sum_{i=1}^{4}\left(\alpha_i+\frac{\beta_i h}{t}+\frac{\gamma_i h^2}{t^2}\right)x(t+(2-i)h)
$$
$$
+h^2 f\left(x(t)+\frac{t-3h}{t}(x(t)-x(t-h))\right),
$$

then simple calculation shows that terms with order less than four will be eliminated if we choose coefficients according to the following equations

$$
\begin{cases}
\alpha_1 = 2 \\
\alpha_2 = -5 \\
\alpha_3 = 4 \\
\alpha_4 = -1
\end{cases},
\qquad
\begin{cases}
\beta_1 = \dfrac{9}{2} - k \\
\beta_2 = -6 + 3k \\
\beta_3 = \dfrac{3}{2} - 3k \\
\beta_4 = k
\end{cases},
\qquad
\begin{cases}
\gamma_1 = m_1 \\
\gamma_2 = -\dfrac{3m_1 + m_2 + 3}{2} \\
\gamma_3 = m_2 \\
\gamma_4 = \dfrac{m_1 - m_2 + 3}{2}
\end{cases},
$$

where $k$, $m_1$, $m_2$ can be chosen randomly. Notice that coefficients of recurrence relation (3.2) satisfy above equations. □

### A.5 Algorithms

---

**Algorithm 4 FISTA**

---

Input: step size $s$

Initial value:  $\boldsymbol{Y}_1 = \boldsymbol{X}_0 \in \boldsymbol{M}_{100}$, $t_1 = 1$.

$\boldsymbol{k}$th iteration ($k \geq 1$). Compute

$$
\boldsymbol{X}_k = \underset{\boldsymbol{X}}{\arg\min} \left\{ \frac{1}{2s} \|\boldsymbol{X} - (\boldsymbol{Y}_k - sg(\boldsymbol{Y}_k))\|^2 + \lambda \|\boldsymbol{X}\|_* \right\},
$$

$$
t_{k+1} = \frac{1 + \sqrt{1 + 4t_k^2}}{2},
$$

$$
\boldsymbol{Y}_{k+1} = \boldsymbol{X}_k + \frac{t_k - 1}{t_{k+1}} (\boldsymbol{X}_k - \boldsymbol{X}_{k-1}).
$$

---

---

**Algorithm 5 Accelerated proximal gradient method**

---

Input: step size $s$

Initial value:  $\boldsymbol{X}_1 = \boldsymbol{X}_0 \in \boldsymbol{M}_{100}$.

$\boldsymbol{k}$th iteration ($k \geq 1$). Compute

$$
\boldsymbol{Y}_k = \boldsymbol{X}_{k-1} + \frac{k-3}{k} (\boldsymbol{X}_{k-1} - \boldsymbol{X}_{k-2}),
$$

$$
\boldsymbol{X}_{k+1} = \underset{\boldsymbol{X}}{\arg\min} \left\{ \frac{1}{2s} \|\boldsymbol{X} - (\boldsymbol{Y}_k - sg(\boldsymbol{Y}_k))\|^2 + \lambda \|\boldsymbol{X}\|_* \right\}.
$$

---

### A.6 Details about Numerical Experiments in Section 4

Here we produce some details for our numerical experiments in Section 4.

Our experiments are taken on a simulated data set. Firstly, we generated the 'true' low rank matrix $\boldsymbol{M}$. To do this, we generate a random matrix $\boldsymbol{M_0}$. Entries of $\boldsymbol{M_0}$ are independent and uniformly distributed on $(0, 20)$. Then we compute the singular value decomposition of $\boldsymbol{M_0}$, that is, $\boldsymbol{M_0} = \boldsymbol{U\Sigma V}^\mathsf{T}$. After that, we set $\boldsymbol{M} = \boldsymbol{U\Sigma_0 V}^\mathsf{T}$, where $\boldsymbol{\Sigma_0}$ is a diagonal matrix with only three nonzero diagonal elements. It is not difficult to prove that $\boldsymbol{M}$ has rank 3.

Secondly, we generate the observation set. For every row of $M$, we choose randomly ten entrys to be observed. As a consequence, 10% entries are observed in total.

After data generation step, we apply the abovementioned algorithms (accelerated proximal gradient method, FISTA and our modified FISTA) with fixed step sizes and backtracking to this data set. The parameter of the loss function (4.1) is $\lambda = 0.005$. For initial point, we simply choose the zero matrix (every entry equals to zero). For backtracking, we set the initial step size as 10 and the decay factor $\beta = 0.1$.

