# OpenReview forum: "A new accelerated gradient method inspired by continuous-time perspective"
_ICLR.cc/2021/Conference — Reject_

### Official Review · AnonReviewer2 · 2020-10-27
**Review on "A new accelerated gradient method inspired by continuous-time perspective"**

**Rating:** 4
**Confidence:** 4

**Review:**

In this paper the authors study a version of accelerated gradient method. Inspired by the ODE analysis of Nesterov accelerated gradient method by Su et.al., the authors propose a different discretization of the ODE by Su et al. The truncation order of this scheme is of a higher order, thus the authors claim that the proposed algorithm is more stable and, therefore,  will converge with larger steps. Unfortunately, I found these statements to be vague.

Apart from the above-mentioned truncation error, the only evidence we have is some simple $2$-dimensional experiment. I believe it is not sufficient. Second, for a new scheme convergence of iterates $(x_n)$ to a solution and the convergence rate $F(x_n) - F(x_*)$ should be proven explicitly, they do not follow automatically. Ideally, we need both sound theory and good experiments to claim that one method is better than another. I am afraid both are missing in this work. The same was done for the modified version of FISTA, where the authors add regularizer without any discussion about convergence of the scheme.

Based on this, I cannot recommend this paper.


I suggest the authors to address the above-mentioned concerns in their revision. I think it would be great if one can show directly the connection between the discretization truncation error and better algorithm performance. Note that, however, already Nesterov's methods have optimal performance. Probably, a significant experimental evidence will help here.

---

> ### Author Response · Authors · 2020-11-18
> **Response to AnonReviewer2**
>
> We thank you for your thought-provoking comments. We respond to your questions and comments in detail below.
> >Apart from the above-mentioned truncation error, the only evidence we have is some simple $2$-dimensional experiment.
>
> In terms of numerical examples, our experiment about matrix completion problem also verifies the better stability of the new method. We would like to point out our findings:
> * For fixed step size, the new method can work with larger step size and so converges faster.
> * For backtracking, the step size is shrunk automatically. We find the final step size of the new method is larger, which confirms that the new method can work with larger step size.
>
> In terms of theoretical guarantee, we have revised our paper by adding a part (subsection 3.3 Absolute stability of Nesterov’s method and the new method) to theoretically explain the better stability of our new method. In this subsection, we use absolute stability theory to analysis the two methods and find that the new method has a larger absolutely stable region, which shows that the new method is more stable and guarantee the use of larger step size.
> >Second, for a new scheme convergence of iterates $(x_n)$ to a solution and the convergence rate $F(x_n)-F(x^*)$ should be proven explicitly, they do not follow automatically.
>
> Firstly, we believe that our theory means convergence. We would like to give a short explanation. Dynamic system theory guarantees that the continuous flow $x(t)$ (solution of the differential equation $\ddot{x}+\frac{3}{t}\dot{x}+\nabla F(x)=0$) converges to an optimal point $x^*$ as $t\to\infty$. So for any small $\varepsilon$, we can find a constant $T$ such that for $t\geq T$, $|x(t)-x^*|<\varepsilon$ always satisfies. Theorem 2 shows that $x_n\;(n=T/h)$ converges to $x(T)$ as $h\to\infty$. So for sufficiently small $h$, the distance between $x_n$ and $x^*$ is not larger than $\varepsilon$.
> Secondly, we agree that the convergence rate of the new method is important. Since the performance of our new method is better, we believe that our method also achieves the optimal convergence rate $\mathcal{O}(n^{-2})$ as Nesterov's method does. Furthermore, due to existence of a $1/s$ term in the convergence rate of Nesterov's method, we believe that our new method improves the convergence by working with larger step size $s$. We leave this point for further research.
>
> Thank you again for your suggestions, and we would appreciate further feedback.

---

> > ### Comment · AnonReviewer2 · 2020-11-24
> > **Response**
> >
> > Thanks authors for their input.
> >
> > Unfortunately, heuristic statements is not a rigorous theory.
> >
> > "For fixed step size, the new method can work with larger step size and so converges faster." - this is not necessary true. Take standard gradient descent. It is easy to construct an example where it will be much slower with a step $1.99/L$ (theoretical upper bound) than $1/L$.
> >
> > "our theory means convergence." - no, your theory only means convergence when step size tends to zero. But this is not what you want to use in practice. All motivation was about using even larger steps than Nesterov's method can.
> >
> > "Since the performance of our new method is better" - it is not better, only for some problem.

---

### Official Review · AnonReviewer4 · 2020-10-28
**Review on "A new accelerated gradient method inspired by continuous-time perspective"**

**Rating:** 4
**Confidence:** 3

**Review:**

UPDATE: After reading through all other reviews and responses by the authors, I share the concern that the theoretical justification of the paper is lacking as the connection between the truncation error and the improved algorithm performance is not rigorously proven. Therefore, I have reduced my score.


Summary:


The paper studies the well-known Nesterov's accelerated gradient method and shows the rate of convergence to the solution of an ordinary differential equation recently proposed by [Su et al, 2014]. Motivated by the proof, the authors then derive a new accelerated method with a faster rate of convergence than the original Nesterov's method, which is shown to be more stable than the original Nesterov's method when the step size is large. The method is combined with the proximal operator into a new algorithm referred to as modified FISTA, which is then applied to the matrix completion problem.


Strengths:

- The paper proves the convergence rate of Nesterov's method to the ODE proposed by [Su et al, 2014]. This proof then motivates them to derive a new faster accelerated method where the truncation error has a higher order of $O(h^4)$ compared to $O(h^3)$ in case of Nesterov's method.
- It is shown in two simple examples that the new method is more stable as it can work with larger step sizes.
- The method is applied to a matrix completion problem, where it is shown to have faster convergence than standard FISTA and Nesterov's gradient method.


Concerns:

- In Section 2.2., the purpose of Lemma 1 and Lemma 2 is not clear without looking into the proof of Theorem 2 in the supplementary material. The flow of the paper could be improved if an intuition was given of which role they play in the proof of Theorem 2.
- Similarly, to understand the motivation for the derivation of the new accelerated method in Section 3, one is required to look at the proof of the convergence in the supplement. Also here it would help to provide a detailed motivation for the derivations already in Section 3.
- In the numerical results in Figure 1, the gap $|F(x_n) − F(x^*)|$ (y-axis) does not seem to monotonically decrease but jump up and down erratically. Also there are periodic wave-like patterns visible in the plot. Why do we see those patterns?
- The resulting accelerated numerical method is never explicitly written down, only the specific version derived for the matrix completion problems. The paper would be better understandable if the general numerical scheme (accelerated method) was written down in form of an algorithm after Section 3.
- In the end of Section 4, a reference to Algorithm 2 is missing. Moreover, Figure 2 and Figure 3 are never referenced in the text.
- Page 2, after (1.2) -> "achive" -> "achieve"


Conclusion:

The proposed method provides a theoretical contribution to the understanding of Nesterov's accelerated gradient method. Moreover, a novel algorithm is proposed which is shown to have a faster convergence to the underlying ODE. In the paper this is shown only for a matrix completion problem but I feel that this new algorithm could be adopted by the community if further experiments prove its worth. On the other hand, the flow and presentation of the paper could be improved. Overall this is a borderline paper but its merits may outweigh its flaws.

---

> ### Author Response · Authors · 2020-11-18
> **Response to AnonReviewer4**
>
> We thank you for your thorough evaluation and positive feedback.  We respond to your questions and comments in detail below.
> >In Section 2.2., the purpose of Lemma 1 and Lemma 2 is not clear without looking into the proof of Theorem 2 in the supplementary material. The flow of the paper could be improved if an intuition was given of which role they play in the proof of Theorem 2. Similarly, to understand the motivation for the derivation of the new accelerated method in Section 3, one is required to look at the proof of the convergence in the supplement. Also here it would help to provide a detailed motivation for the derivations already in Section 3.
>
> Thank you for your suggestion, and we have revised our paper accordingly. Lemma 1 and Lemma 2 are in fact technical, so we move this part to appendix to highlight our main Theorem 2. Moreover, we add some explanation of the motivation for the new method at the beginning of subsection 3.1.
> >In the numerical results in Figure 1, the gap $|F(x_n)-F(x^*)|$ (y-axis) does not seem to monotonically decrease but jump up and down erratically. Also there are periodic wave-like patterns visible in the plot. Why do we see those patterns?
>
> We notice that there exists monotonic version of Nesterov's method, but the original version used in our paper can not guarantee monotonic decrease.
> >The resulting accelerated numerical method is never explicitly written down, only the specific version derived for the matrix completion problems. The paper would be better understandable if the general numerical scheme (accelerated method) was written down in form of an algorithm after Section 3. In the end of Section 4, a reference to Algorithm 2 is missing. Moreover, Figure 2 and Figure 3 are never referenced in the text. Page 2, after (1.2) -> "achive" -> "achieve"
>
> Thank you for you detailed comments! We have corrected these mistakes in our new version accordingly. Specifically, we present general version of our new method as Algorithm 1 at the end of subsection 3.1.
>
> Besides, we would like to introduce our newly added subsection(subsection 3.3 Absolute stability of Nesterov’s method and the new method) to you. In this subsection, we use absolute stability theory to analysis the two methods and find that the new method has a larger absolutely stable region, which shows that the new method is more stable and guarantee the use of larger step size. We think that this subsection completes the theoretical analysis of the better performance of the new algorithm.
>
> Thanks again for the detailed review and we would appreciate further feedback.

---

> > ### Comment · AnonReviewer4 · 2020-11-24
> > **Response**
> >
> > I would like to thank the authors for their detailed response. In the revised version, the authors did some improvements to the flow of the paper as suggested in my review. However, after reading through all other reviews and responses by the authors, I now share the concern by the other reviewers that the theoretical justification of the paper is lacking as the connection between the truncation error and the improved algorithm performance is not rigorously proven. Therefore, I will reduce my score.

---

### Official Review · AnonReviewer3 · 2020-10-29

**Rating:** 4
**Confidence:** 4

**Review:**

summary:
This paper proposes an accelerated method that has a high-order truncation error $O(h^4)$ to the ordinary differential equation $\ddot{x} + \frac{3}{t}\dot{x} + f(x) = 0$ obtained from Nesterov's accelerated method by (Su et al., 2014), while Nesterov's method has $O(h^3)$ error. This implies that the iterates of the proposed method converge to the trajectory of the differential equation faster than those of Nesterov's method. The two toy numerical experiments illustrate such phenomenon for certain large step size. A matrix completion problem experiment is further included.

strong point:
- Finding a method that has a high-order truncation error seems new and interesting, and numerical experiment suggests that such method performs better.

weak points:
- There is no theoretical guarantee on the convergence (rate) to a solution of an optimization problem.
- The reason why we care "large" step sizes seem insufficient, while Nesterov's method is stable for normal step sizes (e.g., $1/L$).
- It is not clear when the step size is considered large, other than using an exhaustive search. Unlike Nesterov's method, the interval of step sizes that guarantee convergence to a solution is not known for the proposed method.
- Numerical experiments are limited.

minor comments:
- page 1: $||F(x_n) - F(x^*)|| \to F(x_n) - F(x^*)$
- page 6: what is the Lipschitz constant for this experiment?
- a figure of two-dimensional toy example could help better illustrate the effect of the truncation error.

---

> ### Author Response · Authors · 2020-11-18
> **Response to AnonReviewer3**
>
> We thank you for your time and your thoughtful review. We respond to your questions and comments in detail below.
> >There is no theoretical guarantee on the convergence (rate) to a solution of an optimization problem.
>
> Firstly, we believe that our theory means convergence. We would like to give a short explanation. Dynamic system theory guarantees that the continuous flow $x(t)$ (solution of the differential equation $\ddot{x}+\frac{3}{t}\dot{x}+\nabla F(x)=0$) converges to an optimal point $x^*$ as $t\to\infty$. So for any small $\varepsilon$, we can find a constant $T$ such that for $t\geq T$, $|x(t)-x^*|<\varepsilon$ always satisfies. Theorem 2 shows that $x_n\;(n=T/h)$ converges to $x(T)$ as $h\to\infty$. So for sufficiently small $h$, the distance between $x_n$ and $x^*$ is not larger than $\varepsilon$.
> Secondly, we agree that the convergence rate of the new method is important. Since the performance of our new method is better, we believe that our method also achieves the optimal convergence rate $\mathcal{O}(n^{-2})$ as Nesterov's method does. Furthermore, due to existence of a $1/s$ term in the convergence rate of Nesterov's method, we believe that our new method improves the convergence by working with larger step size $s$. We leave this point for further research.
> >The reason why we care "large" step sizes seem insufficient, while Nesterov's method is stable for normal step sizes (e.g., $1/L$).
>
> Firstly, we would like to explain why we care large step size. We prove that Nesterov's method and our new method converge to an optimal solution $x^*$ along the same flow $x(t)$. Therefore, larger step size (i.e. better stability) implies faster convergence. This is the reason of better performance of our new method in experiments.
> Besides, we agree that $1/L$ step size ensures the stability of Nesterov's method, which is an important result. In our newly added subsection 3.3, we prove the absolutely stable region of Nesterov's method is $s\cdot\nabla^2F\in[0,4/3]$. Actually, the bound of $\nabla^2F$ is Lipschitz constant $L$, so our theoretical result is approximately coincides with classic choice of step size $1/L$. Furthermore, we prove theoretically that the absolutely stable region of our new method is three times as large as Nesterov's method, which theoretically confirms the choice of larger step size.
> >It is not clear when the step size is considered large, other than using an exhaustive search. Unlike Nesterov's method, the interval of step sizes that guarantee convergence to a solution is not known for the proposed method.
>
> We have revised our paper and added a part (subsection 3.3 Absolute stability of Nesterov’s method and the new method) to theoretically explain the better stability of our new method. In this part, we prove that the absolutely stable region of the new method is $s\cdot\nabla^2F\in[0,4]$, where $s$ denotes the step size. This is a theoretical result about choosing step size. We would like to emphasize that the absolutely stable region of Nesterov's method is $s\cdot\nabla^2F\in[0,4/3]$. Absolutely stable region of our new method is three times as large as that of Nesterov's method, which theoretically confirms the better stability of the new method.
> >page 6: what is the Lipschitz constant for this experiment?
>
> The objective function in matrix completion problem is not smooth, so we can not define $\nabla F$ and there is no Lipschitz constant as previous theoretical analysis.
>
> Thanks again for the detailed review and we would appreciate further feedback.

---

> > ### Comment · AnonReviewer3 · 2020-11-24
> > **Response**
> >
> > Thank you for the response.
> >
> > - I still don't agree that the theory in the paper implies convergence of the proposed method to the optimal solution. Your theory works when $h$ goes to $0$, while your main claim is that the proposed method works well for large $h$.
> > - I appreciate the added section, but the paper still does not have enough numerical and theoretical support to conjecture that the proposed method converges and will have the rate $O(1/n^2)$.
> > - I was asking the Lipschitz constant for the smooth part.
> >
> > For these reasons, I maintain my scores.

---

> > > ### Author Response · Authors · 2020-11-24
> > > **Answer of the third question**
> > >
> > > Thank you for your feedback, and I would like to answer the third question (Lipschitz constant) here.
> > >
> > > The smooth part of the objective function is $G(X)=\frac{1}{2}\|X_{obs}-M_{obs}\|^2$.
> > >
> > > Firstly, we have partial derivative $\frac{\partial G}{\partial x_{ij}}=x_{ij}-m_{ij}$ if the $(i,j)$ component is observed and $\frac{\partial G}{\partial x_{ij}}=0$ if not.
> > >
> > > If we regard $m\times m$ matrix $X$ as a vector with length $m^2$, then the gradient of this part is a vector with same length. Then the distance between $\nabla G(X)$ and $\nabla G(Y)$ is $$\|\nabla G(X)-\nabla G(Y)\|=\sqrt{\sum_{(i,j)\in obs}(x_{ij}-y_{ij})^2}.$$
> > >
> > > Secondly, if we choose Frobenius norm accordingly, then $$\|X-Y\|=\sqrt{\sum_{(i,j)}(x_{ij}-y_{ij})^2}.$$ As a result, we can choose Lipschitz constant $L=1$.
> > >
> > > Thank you again for your comment.

---

### Official Review · AnonReviewer1 · 2020-10-30
**Review for "A new accelerated gradient method inspired by continuous-time perspective"**

**Rating:** 4
**Confidence:** 3

**Review:**

Review:  This paper refines the the truncation error analysis for discretizing the ODE to obtain accelerated optimization method.  The truncation results include higher order term. Built upon the analysis, the authors propose a new method which is claimed to be more stable for large step size and converges faster. Numerical evidence on matrix completion problem is provided.


Pros:

+ The truncation error analysis is new.

+ Overall, the paper is clearly written.



Cons:

- The biggest concern that I have with the paper is that it is unclear to me whether the convergence rate is really improved or not. From my understanding, truncation error is different from the convergence rate. What does Theorem 3 really imply here? It seems to me that Theorem 3 does not guarantee that there is an improvement in the convergence rate. A rigorous quantification of the convergence rate needs to be provided to justify the claim "the proposed method converges faster."


- Even the claim on "stability" is not well justified. Two simple examples do not provide that much evidence here.

- This paper does not provide enough details such that the numerical results can be easily reproduced.




Suggestions for improvements:
 It will significantly strengthen the paper if the authors can provide more theoretical justifications for the claim that their proposed method is faster and more stable. It is also important to clarify the true implications of the truncation error analysis on the algorithm performance.

---

> ### Author Response · Authors · 2020-11-18
> **Response to AnonReviewer1**
>
> We thank you for your time and your thoughtful comments on our work. We have revised our paper by adding a part (subsection 3.3 Absolute stability of Nesterov’s method and the new method) to theoretically explain the better stability of our new method. In this subsection, we use absolute stability theory to analysis the two methods and conclude that the new method has a larger absolutely stable region, which shows that the new method is more stable and guarantee the use of larger step size. We respond to your questions and comments in detail below.
> >The biggest concern that I have with the paper is that it is unclear to me whether the convergence rate is really improved or not. From my understanding, truncation error is different from the convergence rate. What does Theorem 3 really imply here? It seems to me that Theorem 3 does not guarantee that there is an improvement in the convergence rate. A rigorous quantification of the convergence rate needs to be provided to justify the claim "the proposed method converges faster."
>
> The algorithm performance is improved because our new algorithm can work with larger step size when converges to one optimal point $x^*$ along the continuous flow $x(t)$ which is also the limit flow of Nesterov's method. As a result, our new method is more stable and converges faster. We give a theoretical proof of the chosen of larger step size via absolutely stable theory in subsection 3.3 ($4/3$ for Nesterov's method and $4$ for our new method). Besides, numerical experiments confirm our theoretical finding.
> We agree that the convergence rate of the new method is important. Since the performance of our new method is better, we believe that our method also achieves the optimal convergence rate $\mathcal{O}(n^{-2})$ as Nesterov's method does. Furthermore, due to existence of a $1/s$ term in the convergence rate of Nesterov's method, we believe that our new method improves the convergence by working with larger step size $s$. We leave this point for further research.
> >Even the claim on "stability" is not well justified. Two simple examples do not provide that much evidence here.
>
> We would like to explain that the concept "stability" in our paper means that the algorithm can work well with large step size, instead of diverge. We have added a new subsection to verify the better stability of the new method via absolute stability theory. In terms of numerical examples, our experiment about matrix completion problem also verifies the better stability of the new method. We would like to point out our findings:
> * For fixed step size, the new method can work with larger step size and so converges faster.
> * For backtracking, the step size is shrunk, which confirms that the new method can work with larger step size.
>
> >This paper does not provide enough details such that the numerical results can be easily reproduced.
>
> We have added some details of our experiments in appendix (A.6 Details about Numerical Experiments in Section 4) and uploaded the codes of experiments.
> >It is also important to clarify the true implications of the truncation error analysis on the algorithm performance.
>
> Thank you for your insightful suggestion. We agree that this is the core issue of our paper and related works. In this paper, we find that algorithm with higher truncation order will converge to the continuous limit faster, thus is more stable (subsection 3.2). We think that our explanation can partly answer this question, but a precise solution needs further work.
>
> Thanks again for the detailed review and we would appreciate further feedback.

---

### Author Response · Authors · 2020-11-24
**The end of the discussion phase approaching**

Dear reviewers,

Could you please go over our responses and the revision since we can have interactions with you only by today (24th)? We have responded to you comments and faithfully reflected them in the revision, and provided additional analysis that you have required. We sincerely thank you for your time and efforts in reviewing our paper, and your insightful and constructive comments.

Thanks, Authors

---

### Decision · Program_Chairs · 2021-01-07
**Final Decision**

**Decision:**

Reject

**Comment:**

The paper studies a high-order discretization of the ODE corresponding to Nesterov's accelerated method, as introduced by Su-Boyd-Candes. The main claim of the paper is that the more complex discretization scheme leads to a method that is more stable and faster. However, the theoretical claims do not seem sufficiently supported, and the experimental results are insufficient to judge the usefulness of the proposed approach. Thus, the reviews could not recommend acceptance, and I concur. The authors are advised to revise the paper to provide more theoretical and experimental evidence for usefulness/competitiveness of the proposed approach, and resubmit to a different venue.